# Cu-catalyzed asymmetric regiodivergent electrosynthesis and its application in the enantioselective total synthesis of (-)-fumimycin

Tian Xie[1], Jianming Huang[2], Juan Li[2], Lingzi Peng[1], Jin Song[2] & Chang Guo [1]✉

Quaternary amino acids are one of the essential building blocks and precursors of medicinally important compounds. Various synthetic strategies towards their synthesis have been reported. On the other hand, developing core-structure-oriented cross-dehydrogenative coupling (CDC) reactions, is a largely unsolved problem. Herein, we describe a copper-catalyzed regiodivergent electrochemical CDC reaction of Schiff bases and commercially available hydroquinones to obtain three classes of chiral quaternary amino acid derivatives for the efficient assembly of complex scaffolds with excellent stereocontrol. The electrochemical anodic oxidation process with slow releasing of quinones serves as an internal syringe pump and provides high levels of reaction efficiency and enantiomeric control. The utility of this strategy is highlighted through the synthetic utility in the asymmetric total synthesis of (-)-fumimycin.

Stereochemically rigid α,α-disubstituted amino acids are a prevalent structural feature in a wide range of biologically active compounds, fueling the ongoing development of a powerful and efficient tool for dramatically increasing the complexity of organic molecules. Fumimycin, which was originally isolated from Aspergillus fumisynnematus, has been shown to exhibit antibacterial and against peptide deformylase inhibitory activity (Fig. 1a)[1]. Elegant contributions from the groups of Bräse[2–4], Zhou[5], Roche[6], Piersanti[7], and Watanabe[8] have been dedicated to the development of powerful synthetic strategies for the efficient synthesis of fumimycin. Despite significant progress, the assembly of quaternary carbon stereocenters for asymmetric catalytic synthesis of fumimycin in a highly stereoselective manner is still challenging, most likely due to the congested nature of such scaffolds and a lack of reliable synthetic routes for the appropriate installation of the hydroquinone moiety. In this context, the development of cross-dehydrogenative coupling (CDC) methods[9,10] in a catalytic asymmetric fashion would significantly simplify synthetic access to α-quaternary amino acids, which would be invaluable in delivering compound libraries to benefit chemical space in drug design and screening[11].

Quinone and its derivatives have long served as useful synthetic precursors which allow the facile synthesis of functionalized aromatic rings[12]. Accordingly, a plethora of catalytic enantioselective transformations based on the 1,4-addition to the quinone sp[2] hybridized carbons to access valuable α-arylated products have been developed[13–17]. Very recently, seminal works by the List[18] and Jørgensen[19–21] groups revealed asymmetric α-aryloxylation of cyclic ketones or aldehydes via the 1,6-addition of benzoquinones. Nowadays, electrochemical synthesis has received extensive attention as a sustainable way to replace chemical oxidants and reductants[22–33]. The tunability of the potential and electric current, in particular, provides a unique handle for organic electrochemistry to precisely tame redox transformations and exquisitely control the release of reactive intermediates. As a result, many transformations have been developed based on the electro-generated reactive intermediates[34]. In 2010, Jørgensen and

[1]Hefei National Research Center for Physical Sciences at the Microscale and Department of Chemistry, University of Science and Technology of China, Hefei 230026, China. [2]Institutes of Physical Science and Information Technology, Key Laboratory of Environment-Friendly Polymeric Materials of Anhui Province, Anhui University, Hefei 230601, China. ✉e-mail: guochang@ustc.edu.cn

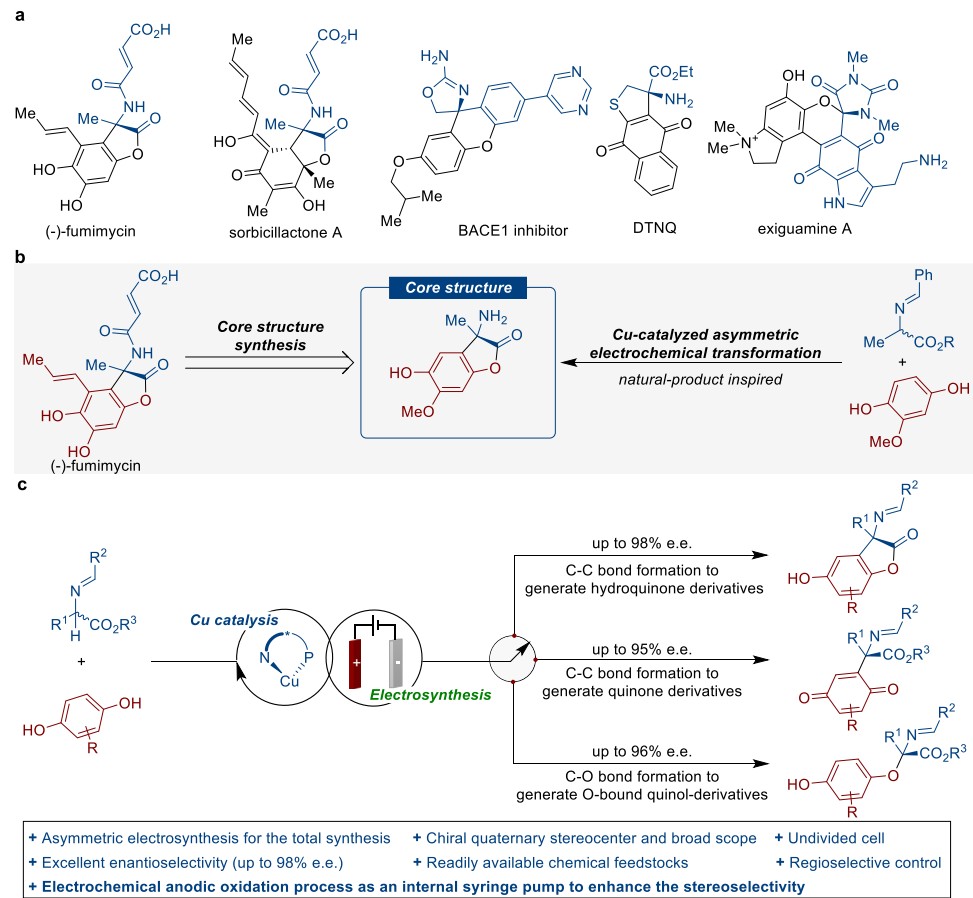

**Fig. 1 | Design of Cu-catalyzed enantioselective electrochemical regiodivergent synthesis and applications in the asymmetric total synthesis of (-)-fumimycin. a** Representative biologically active quaternary amino acids. **b** Retrosynthesis of (-)-fumimycin. **c** Cu-catalyzed highly tunable CDC reactions with C/O selectivity of hydroquinones.

coworkers pioneered the development of enantioselective anodic oxidation/organocatalytic reactions for the α-arylation of aldehydes with in situ generated quinones[35]. On the other hand, the combination of chiral Lewis acid catalysis with the unique reactivity of electrochemically generated intermediates[36–39] has recently shown the potential to offer a valuable platform for the development of a diverse range of stereocontrolled reactions[40–57]. Thus, we envisioned that the integration of two units of intermediates, including electrochemically generated quinone species and Lewis acid-bound nucleophiles, would offer opportunities in switchable hydroquinone transformations to access site-diverse regioisomers.

The retrosynthetic analysis of (-)-fumimycin, which focuses on directly assembling quaternary carbon stereocenters, allows the design of the synthetic strategy that utilizes the enantioselective coupling of Schiff bases with electrochemically generated quinones, and would offer mechanistically diverse transformations to access valuable chiral α-tertiary amino ester derivatives rationally and predictably (Fig. 1b). Furthermore, well-established N-metallated azomethine ylides, serving as a binding cavity, reveal distinct reactivity patterns and enable the diversity-oriented synthesis of unnatural α-amino acids with consistently high enantioselectivity[58,59]. In this work, we describe the chiral copper-catalyzed, highly enantioselective, regiodivergent, electrochemical CDC reactions of Schiff bases with hydroquinone derivatives, providing diverse synthetic routes toward chiral quaternary amino esters with multiple substituent patterns via anodic oxidation and Lewis acid catalysis (Fig. 1c). Notably, the efficiency of this approach is demonstrated by its application to the enantioselective total synthesis of (-)-fumimycin.

## Results and discussion

We initiated the study by investigating the electrochemical reaction between racemic ketimine ester **1a** and 2,3-dimethylhydroquinone **2a** in tetrahydrofuran (THF) (Table 1). The use of chiral box ligand (S,S)-**L1** afforded the 1,4-addition product **3a** with poor stereocontrol (68% yield, 38% e.e., entry 1). We next evaluated a variety of chiral ligands for this asymmetric electrochemical CDC reaction (entries 2-5), and the product **3a** was obtained in 71% yield and 92% e.e. with the use of Phosferrox ligand (S,Sp)-**L5** (entry 5). Interestingly, when the reaction was carried out at −10 °C, a switchable reaction pathway that results in kinetic resolution (KR, entry 6) over dynamic kinetic asymmetric transformation (DyKAT, entry 5) of racemic ketimine ester **1a** successfully afforded **3a** in 57% yield and 92% e.e., as well as the recovery of (S)-**1a** in 30% yield with 98% e.e. (entry 6). Further studies focused on the evaluation of different ester moieties of ketimine esters **1** (entries 7–9), and the hydroquinone adduct **4a**, which can be applied in the analog synthesis of (-)-fumimycin, could be obtained via an initial 1,4-addition and a subsequent intramolecular cyclization. To our surprise, the use of 1-naphthyl ester **1d** instead of methyl ester **1a** resulted in a successful switchable production of the desired hydroquinone product **4a** over **3a**, albeit with 81% e.e. (entry 9). Under the electrolytic conditions, the stereoselectivity was further improved with the use of Phosferrox ligand (S,Sp)-**L4**, leading to the desired product **4a** in 80% yield with 95% e.e. (entry 10). Further studies showed that the hydroquinone backbone was also susceptible to the reaction. We successfully switched the α-arylated process to the aryloxylation reaction using bulky 2,6-dimethylhydroquinone **2b** under otherwise

## Table 1 | Optimization of the reaction conditions

| Entry | L | 1 | 2 | T | Results of 3 | Results of 4a | Results of 5a |
|---|---|---|---|---|---|---|---|
| 1 | (S,S)-**L1** | **1a** | **2a** | 10 °C | 68% yield, 38% e.e. | – | – |
| 2 | (R)-**L2** | **1a** | **2a** | 10 °C | 54% yield, 34% e.e. | – | – |
| 3 | (R)-**L3** | **1a** | **2a** | 10 °C | 52% yield, 3% e.e. | – | – |
| 4 | (S,S_p)-**L4** | **1a** | **2a** | 10 °C | 61% yield, 89% e.e. | – | – |
| 5 | (S,S_p)-**L5** | **1a** | **2a** | 10 °C | 71% yield, 92% e.e. | – | – |
| 6[a] | (S,S_p)-**L5** | **1a** | **2a** | −10 °C | 57% yield, 92% e.e. | – | – |
| 7 | (S,S_p)-**L5** | **1b** | **2a** | −10 °C | 65% yield, 93% e.e. | 16% yield, 97% e.e. | – |
| 8 | (S,S_p)-**L5** | **1c** | **2a** | −10 °C | – | 66% yield, 87% e.e. | – |
| 9 | (S,S_p)-**L5** | **1d** | **2a** | −10 °C | – | 83% yield, 81% e.e. | – |
| 10 | (S,S_p)-**L4** | **1d** | **2a** | −10 °C | – | 80% yield, 95% e.e. | – |
| 11 | (S,S_p)-**L4** | **1d** | **2b** | −10 °C | – | – | 78% yield, 92% e.e. |

Reactions were performed by using racemic ketimine ester **1** (0.15 mmol), hydroquinone **2** (0.225 mmol), Cu(MeCN)$_4$BF$_4$ (10 mol%), **L** (12 mol%), KOAc (0.3 mmol), $^n$Bu$_4$NClO$_4$ (0.07 M), and THF (4 mL) under constant-current conditions in an undivided cell. Enantiomeric excess was analyzed by chiral HPLC.

$^n$Bu$_4$NClO$_4$ tetrabutylammonium perchlorate, 1-Nap 1-naphthyl, Me methyl, Bn benzyl.

[a] **1a** was recovered in 30% yield with 98% e.e.

identical reaction conditions, producing **5a** in the majority (entry 11, 78% yield, 92% e.e.).

### Mechanistic investigations

Some preliminary studies were carried out to better understand the mechanism of copper-catalyzed electrochemical regiodivergent reactions (Fig. 2). Initially, control experiments toward KR and DyKAT of racemic ketimine ester **1** were performed (Fig. 2a). Under conditions shown in Table 1, entry 11, (rac)-**1a** was successfully resolved to its S-enantiomer with considerable stereocontrol (85% e.e., Fig. 2a, entry 1). In contrast, DyKAT of 1-naphthyl ester **1d** with **2b** went smoothly, generating the desired product **5a** in 78% yield and 92% e.e. (entry 2). The reaction involving **1a** and **2b** in a ratio of 2:1 resulted in **5a'** in 54% yield and 89% e.e., with (S)-**1a** recovered in 60% e.e. (entry 3); whereas the coupling of **1d** with **2b** in a ratio of 2:1 led to **5a** in 90% yield and 92% e.e. with racemic **1d** recovered (entry 4). Subsequently, racemization profiles of (S)-**1a** (94% e.e.) and (S)-**1d** (48% e.e.) were investigated in the presence of copper catalytic system and substrate **2b** (Fig. 2b). Rapid racemization of Schiff base (S)-**1d** was observed at −10 °C within 20 min, while (S)-**1a** remained unaffected over 6 h. More importantly, (S)-**1a** (94% e.e.) became racemic within 2 h at 10 °C. From these results, we concluded that both the ester moiety of Schiff base **1** and reaction temperature were responsible for the switchable KR and DyKAT process.

As shown in Fig. 2c, the $^1$H NMR spectrum for the reaction progress of **1d** and **2a** was monitored under standard conditions (Table 1, entry 10). The full conversion of the starting materials was achieved within 5 h, delivering dehydrogenation adduct **4a** smoothly with its e.e. value remaining stable at 95% during the reaction progress. The quinone intermediate **2a'** was not detected until the reaction was nearly complete, which suggested a slow release of quinone intermediate from dimethylhydroquinone **2a**. We further investigated the potential pathway arising from the electrochemical-free process with quinone **2a'** (Fig. 2d). Surprisingly, exposing quinone **2a'** with **1d** in one portion resulted in the formation of **4a** in essentially the same yield, albeit with significantly lower e.e. (Fig. 2d, t = 0 h, 48% e.e.). We speculated that the concentration of quinone would play a key role in this reaction. Thus, the enantiomeric excess of product **4a** was measured as a function of the addition time of quinone **2a'**. Using a syringe pump, **2a'** was lowly added over 4.5 h, and the results of product **4a** (95% e.e.) perfectly matched with the standard electricity-driven protocol (Table 1, entry 10). On this basis, we hypothesized that high levels of reaction efficiency and enantiocontrol can be achieved through an internal syringe pump protocol wherein the controlled release of reactive quinones from the in situ anodic oxidation process of hydroquinones is accomplished.

Taking into account the combined results of mechanistic studies, a proposed mechanism for the Cu-catalyzed regiodivergent CDC

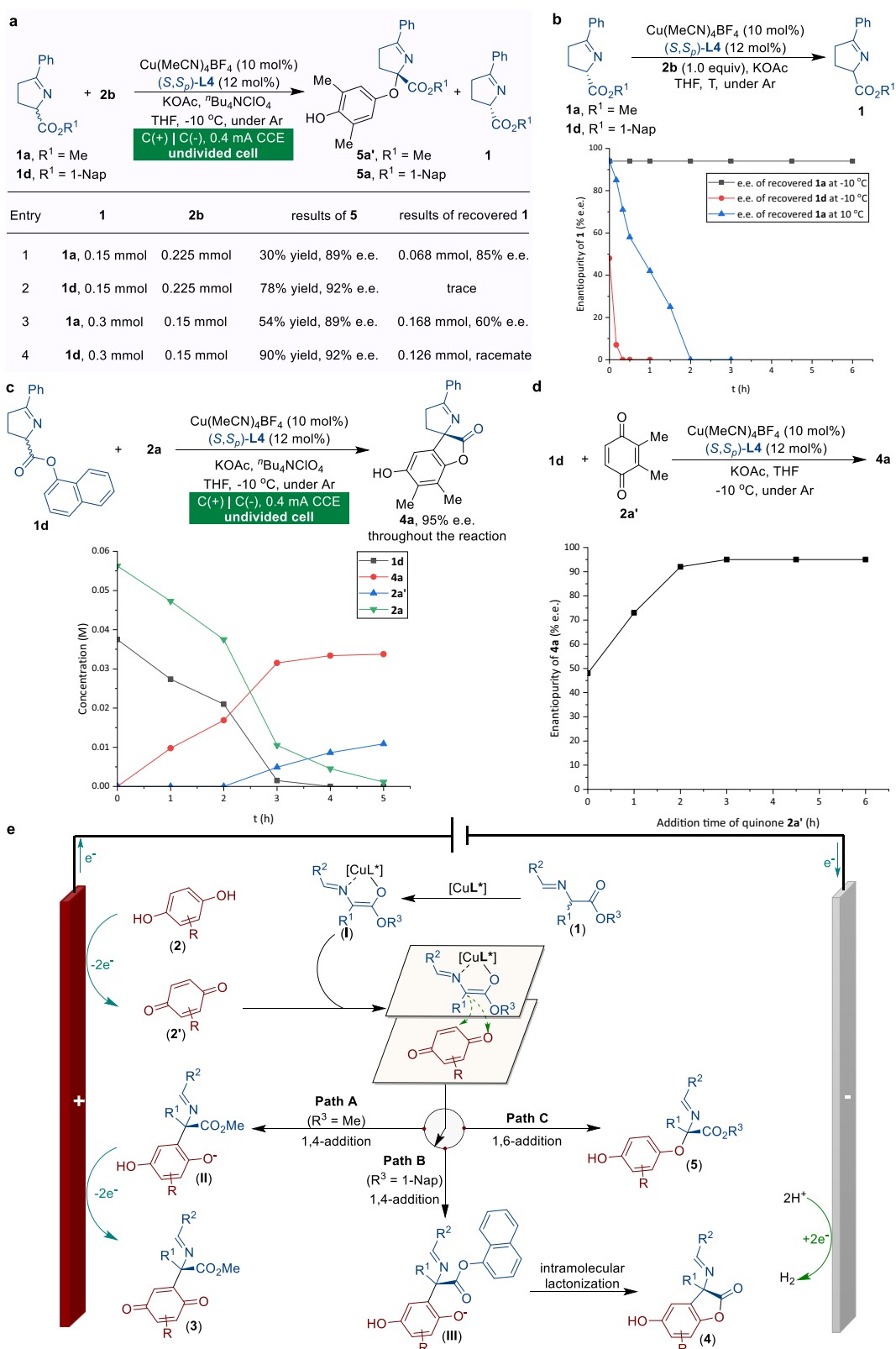

**Fig. 2 | Mechanistic studies. a** Control experiments. **b** Racemization profile of (*S*)-**1a** and (*S*)-**1d**. **c** Kinetic profile of the reaction of **1d** and **2a**. **d** Effect of addition time of quinone **2a'** on the e.e. values of **4a**. **e** Proposed catalytic cycles for the copper-catalyzed enantioselective electrochemical regiodivergent CDC reactions.

reactions is outlined in Fig. 2e. The catalytic cycles begin with the coordination of the copper catalyst to Schiff base **1** to generate the nucleophilic metallated azomethine ylide **I**. Meanwhile, hydroquinone **2** is electrochemically oxidized at the anode to produce quinone species **2'**. Remarkably, mechanistic studies show that an anodic oxidation

process exhibiting as the internal syringe pump to the controllable release of quinone species is essential to the stereoselective transformation. The structure of the hydroquinones employed in the reaction defined the initial selectivity between the competing 1,4-addition and 1,6-addition pathways with Cu-coordinated azomethine ylide **I**.

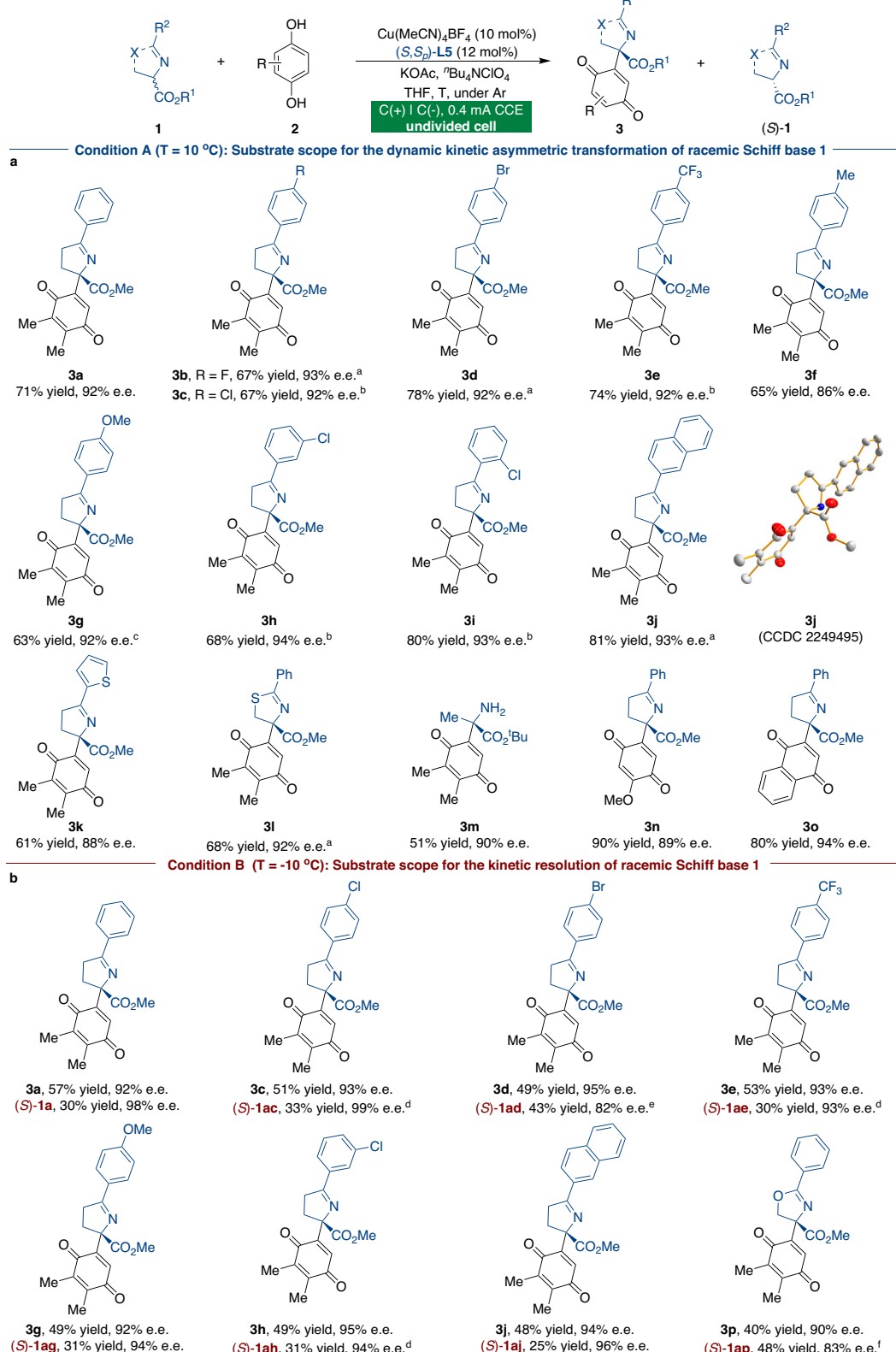

**Fig. 3 | Substrate scope for KR and DyKAT for the synthesis of quinone adducts 3. a** Substrate scope for the DyKAT of **1**. The reactions were carried out using racemic ketimine ester **1** (0.15 mmol), hydroquinone **2** (0.225 mmol), Cu(MeCN)₄BF₄ (10 mol%), (*S,S*ₚ)-**L5** (12 mol%), KOAc (0.3 mmol), ⁿBu₄NClO₄

(0.07 M), and THF (4 mL) under constant-current conditions in an undivided cell. **b** Substrate scope for the kinetic resolution of **1**. ᵃAt 0 °C. ᵇAt −10 °C. ᶜAt 5 °C. ᵈAt −25 °C. ᵉAt −30 °C. ᶠAt 10 °C.

Condensation of **I** with less steric hydroquinone (such as 2,3-dime-thylhydroquinone, **2a**) promotes the 1,4-addition to produce the α-arylated products (**II** or **III**), and the following reaction process is influenced by different ester groups of Schiff base **1**. In other words, methyl ester Schiff base (R³ = Me) would undergo 1,4-addition and electrochemical oxidation to afford quinone product **3** (Path A), whereas naphthyl ester Schiff base (R³ = 1-Nap) is preferred for sub-sequent intramolecular lactonization to afford product **4** (Path B). Using sterically hindered 2,6-disubstituted hydroquinone **2** as a sub-strate, 1,6-addition reactivity prevails to provide the final α-aryloxylation product **5** and regenerate the copper catalyst (Path C).

We investigated the generality of this reaction with different substituted Schiff bases after optimizing the reaction conditions for the synthesis of quinone adduct **3** (Fig. 3). As shown in Fig. 3a, a broad range of racemic ketimine esters **1** gave products with high yields and enantioselectivities. Substrates bearing electron-withdrawing or -donating groups on the aromatic rings participated in the transfor-mation, resulting in excellent enantioselectivities (**3a**-**3i**). Furthermore, 2-naphthyl and heteroaromatic ketimine esters smoothly underwent this catalytic transformation with high enantioselectivities (**3j** and **3k**). The absolute configuration of adduct **3j** was unambiguously assigned by X-ray analysis. Furthermore, methyl 2-thiazoline-4-carboxylate and aldimine ester were effectively converted into the corresponding products (**3l** and **3m**). The generality of the electrochemical CDC reaction in terms of hydroquinone substituents was also explored, and the corresponding products were obtained with satisfactory results (**3n** and **3o**). By lowering the reaction temperature, we were able to expand the purview of this catalytic KR reaction for various racemic

Schiff bases **1** (Fig. 3b). The KR of Schiff base **1** with different benzene ring substituents proceeded smoothly, with nearly 50% conversion and with excellent KR results in all instances. Methyl 2-phenyl-2-oxazoline-4-carboxylate **1ap** also worked well to deliver quinone product **3p** with high levels of stereoselectivity, while unreacted enantiomeric Schiff base (S)-**1ap** was recovered with 83% e.e.

Next, we explored the scope of the tandem annulation reactions of 1-naphthyl ester **1**, for which lactone products **4** are specifically available (Fig. 4). A wide variety of Schiff bases and hydroquinones were investigated under optimized reaction conditions (Table 1, entry 10). Aromatic and heteroaromatic Schiff bases were transformed cat-alytically with good enantioselectivities (**4a**-**4j**). Notably, the opposite enantiomer of **4a** can be obtained by using ligand (R,Rₚ)-**L4** under otherwise identical conditions, and the absolute configuration of *ent*-**4a** was assigned by single-crystal X-ray diffraction analysis. Further exploration showed that naphthalen-1-yl 2-phenyl-4,5-dihydrothiazole-4-carboxylate was also an appropriate substrate (**4k**). We then attempted to assess the applicability of this methodology to other types of hydroquinones. The reaction with different substituents on the hydroquinone ring went smoothly, obtaining moderate yields and good enantioselectivities (**4l**-**4o**).

In comparison to the prevalence of reactions involving α-arylated reactivity, asymmetric α-aryloxylation for C−O bond formation has received far less attention. A wide range of Schiff bases **1** was investi-gated with 2,6-disubstituted hydroquinones (Fig. 5a). Substitutions on the benzene ring of Schiff bases had little effect on the reaction (**5a**-**5f**). The absolute configuration of **5b** was assigned by single-crystal X-ray diffraction analysis. Furthermore, variations in the different

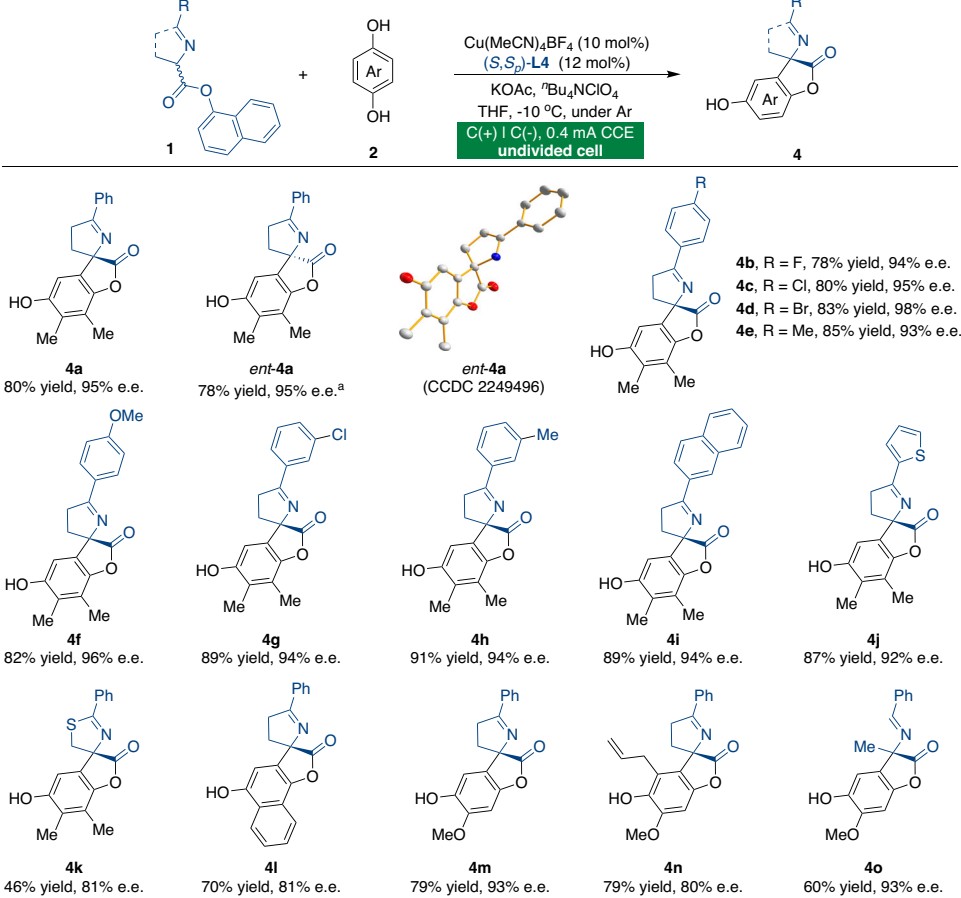

**Fig. 4 | Substrate scope for the tandem annulation reactions.** Unless otherwise specified, all the reactions were carried out using racemic ketimine ester **1** (0.15 mmol), hydroquinone **2** (0.225 mmol), Cu(MeCN)₄BF₄ (10 mol%), (S,Sₚ)-**L4** (12 mol%), KOAc (0.3 mmol), ⁿBu₄NClO₄ (0.07 M), and THF (4 mL) at −10 °C under constant-current conditions in an undivided cell. ᵃ(R,Rₚ)-**L4** was used.

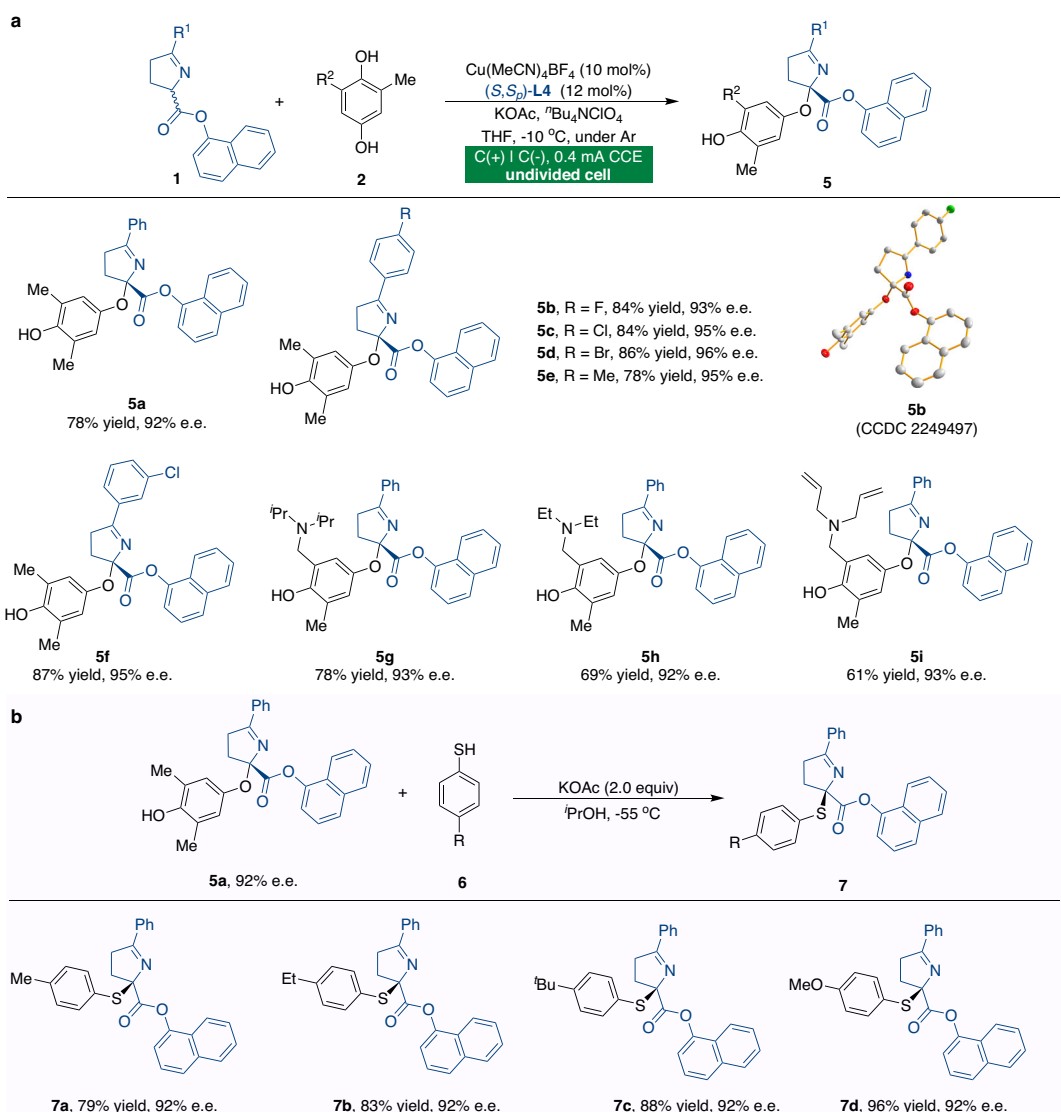

**Fig. 5 | Substrate scope for the asymmetric α-aryloxylation reactions and synthetic transformations. a** Isolated *O*-bound quinol-derivatives. Unless otherwise specified, all the reactions were carried out using racemic ketimine ester **1** (0.15 mmol), hydroquinone **2** (0.225 mmol), Cu(MeCN)$_4$BF$_4$ (10 mol%), (*S*,*S$_p$*)-**L4** (12 mol%), KOAc (0.3 mmol), $^n$Bu$_4$NClO$_4$ (0.07 M), and THF (4 mL) at −10 °C under constant-current conditions in an undivided cell. **b** Synthesis of optically active thioesters.

substituted groups of hydroquinones **2** had no discernible effect on the reaction outcomes, giving the corresponding α-aryloxylation products in good yields and enantioselectivities (**5g**-**5i**). To test the synthetic utility of the electrochemical CDC reactions, optically active α-aryloxylation product **5a** can be readily elaborated (Fig. 5b). Under basic conditions, the treatment of compound **5a** with different thiols **6** furnished a variety of valuable quaternary thioethers **7** in high yields with maintaining e.e. (**7a**-**7d**).

The utility of the current method is further demonstrated by the enantioselective total synthesis of (-)-fumimycin with high stereochemical purity (Fig. 6). Accordingly, **4o** was treated with allyl bromide and potassium carbonate to produce allyl ether **8** in 80% yield and 93% e.e. Acidic hydrolysis of the imine group, followed by the treatment with *tert*-butyl protected acid chloride, generates the corresponding amide **9** without affecting the enantioselectivity. The thermal Claisen rearrangement was performed to afford **10** in DMF at 180 °C[2, 3]. Subsequent isomerization of the terminal double bond was accomplished under rhodium catalysis, and the desired product **11** was afforded without loss of enantioselectivity. Finally, the methyl group and the *tert*-butyl ester were cleaved to yield (-)-fumimycin successfully. All the

spectroscopic data for our synthetic product are in agreement with those reported in the literature[4, 7]. Notably, the absolute configuration of (-)-fumimycin is consistent with the X-ray diffraction analysis of **9**.

We have developed a Cu-catalyzed asymmetric electrochemical regiodivergent CDC reaction of Schiff bases and hydroquinones. This methodology enables efficient and economic access to a regiodivergent and enantioselective synthesis of valuable amino esters bearing quaternary stereocenters with high functional group tolerance. The regioselectivity could be switched by using the appropriate substrates, and the electrochemical process acts as an internal syringe pump to boost the stereoselectivity, offering an effective platform for future efforts to create complex and value-added quaternary amino acids. The synthetic utility of this asymmetric electrochemical methodology is demonstrated by its application to the total synthesis of (-)-fumimycin.

## Methods
### General procedure for the synthesis of chiral products 3
In a Schlenk tube equipped with two carbon electrodes, Cu(MeCN)$_4$BF$_4$ (0.015 mmol, 10 mol%) and (*S, S$_p$*)-**L5** (0.018 mmol,

**Fig. 6 | Asymmetric total synthesis of (-)-fumimycin.** The current methodology could allow for the enantioselective product of (-)-fumimycin. The absolute structure of (-)-fumimycin agrees with the X-ray diffraction investigation of **9**.

12 mol%) were stirred in anhydrous THF (2 mL) under argon at 25 °C for 10 min. **1** (0.15 mmol), **2** (0.225 mmol), $^n$Bu$_4$NClO$_4$ (0.28 mmol), KOAc (0.3 mmol), and THF (2 mL) were added successively under argon. The constant current (I = 0.4 mA) electrolysis was carried out at 10 °C until complete consumption of the substrate (monitored by TLC, 8–12 h). The solvent was removed under reduced pressure. The residue was purified by silica gel chromatography to afford the desired product **3**.

### General procedure for the synthesis of chiral products 4

In a Schlenk tube equipped with two carbon electrodes, Cu(MeCN)$_4$BF$_4$ (0.015 mmol, 10 mol%) and (*S,Sp*)-**L4** (0.018 mmol, 12 mol%) were stirred in anhydrous THF (2 mL) under argon at 25 °C for 10 min. **1** (0.15 mmol), **2** (0.225 mmol), $^n$Bu$_4$NClO$^4$ (0.28 mmol), KOAc (0.3 mmol), and THF (2 mL) were added successively under argon. The constant current (I = 0.4 mA) electrolysis was carried out at −10 °C until complete consumption of the substrate (monitored by TLC, 6–8 h). The solvent was removed under reduced pressure. The residue was purified by silica gel chromatography to afford the desired product **4**.

### General procedure for the synthesis of chiral products 5

In a Schlenk tube equipped with two carbon electrodes, Cu(MeCN)$_4$BF$_4$ (0.015 mmol, 10 mol%) and (*S,Sp*)-**L4** (0.018 mmol, 12 mol%) were stirred in anhydrous THF (2 mL) under argon at 25 °C for 10 min. **1** (0.15 mmol), **2** (0.225 mmol), $^n$Bu$_4$NClO$_4$ (0.28 mmol), KOAc (0.3 mmol), and THF (2 mL) were added successively under argon. The constant current (I = 0.4 mA) electrolysis was carried out at −10 °C until complete consumption of the substrate (monitored by TLC, 8–12 h). The solvent was removed under reduced pressure. The residue was purified by silica gel chromatography to afford the desired product **5**.

### Data availability

Crystallographic data for the structures reported in this article have been deposited at the Cambridge Crystallographic Data Centre under deposition numbers CCDC 2249495 (**3j**), CCDC 2249496 (*ent*-**4a**), CCDC 2249497 (**5b**), and CCDC 2249498 (**9**). Copies of the data can be obtained free of charge via https://www.ccdc.cam.ac.uk/structures/. All other data supporting the findings of this study, including experimental procedures and compound characterization, NMR, and HPLC are available within the Article and its Supplementary Information or from the authors. NMR data in a mnova file format and HPLC traces are available at Zenodo at https://zenodo.org/record/8249500, under the Creative Commons Attribution 4.0 International license.

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

## Acknowledgements
The authors acknowledge financial support from the National Natural
Science Foundation of China (grant no. 21971227, 22222113), CAS Project
for Young Scientists in Basic Research (YSBR-054), and the Fundamental
Research Funds for the Central Universities (WK9990000090,
WK9990000111).

## Author contributions
C.G. conceived the project. T.X. performed the experiments and ana-
lyzed the data. J.H., J.L. and L.P. synthesized some of the substrates and
ligands. C.G. and J.S. wrote the paper. All authors discussed the results
and commented on the manuscript.

## Competing interests
The authors declare no competing interests.

## Additional information
**Supplementary information** The online version contains
supplementary material available at

Chang Guo.

**Peer review information** *Nature Communications* thanks the anon-
ymous reviewers for their contribution to the peer review of this work. A
peer review file is available.

