## [Peer Review File · Nature Communications]

Cu-catalyzed asymmetric regiodivergent electrosynthesis and its application in the enantioselective total synthesis of (-)-fumimycinReviewers' Comments:

Reviewer #1:

Remarks to the Author:

Organic electrochemistry has emerged as a powerful tool for organic synthesis. This paper by Guo et al describes an intriguing electrochemical regiodivergent synthesis of valuable amino acid derivatives bearing a quaternary stereocenter in high yields and enantioselectivities. The combination of electrochemistry and Lewis acid catalysis for asymmetric transformation is the key to the achievement of three classes of transformations, based on which asymmetric synthesis of (-)-fumimycin is demonstrated feasible via a concise route. In addition, the mechanistic studies add additional value to the understanding of chemistry. Based on the high challenges of prepared products and the high synthetic value of this methodology, I strongly support the publication of this work in Nature Communications after addressing following minor issues.

There are a few minor points to be considered:

- (1) In Fig. 3a, such as product 3n, or in Fig. 4 such as product 4m and 4o, I wondered if the authors observed any regioisomers.
- (2) For Table 1: some abbreviations should be added to the footnote, such as Nap.
- (3) Page 11, change "in the literatures" to "in the literature".
- (4) A related reference on asymmetric electrochemical transformation is suggested to be cited: Chin. J. Org. Chem. 2020, 40, 3738-3747.
- (5) The SI is well prepared and written with good quality of spectra. The eluting solvents for chromatography are recommended to be provided.

Reviewer #2:

Remarks to the Author:

Guo and co-workers report an electrochemical oxidation to generate quinones followed by copper-catalyzed asymmetric addition reactions. Upon alteration of reaction conditions, three types of hydroquinone transformations were realized. My first impression on this manuscript is that it reports a nice work. However, after the literature survey, this referee believes this work doesn't represent an important advance towards asymmetric transformation of hydroquinone.

First, the electrochemical oxidation of hydroquinone to quinone is a well demonstrated chemistry. Even for the electrochemical utilization of in situ generated quinones for asymmetric reactions, many beautiful works have been reported (a seminal work: *Angew. Chem. Int. Ed.* 2009, 10.1002/anie.200904754). Therefore, the chemistry reported herein lacks enough novelty.

Second, the asymmetric transformation of quinone has been realized by many groups (ref. 13-21 and others). In the previous reports, excellent enantioselectivities could be achieved. However, in the present work, most of the ee values are below 94%.

Overall, the main achievement of this work is the use of electrons as chemical oxidant alternatives to generate quinones from hydroquinones, while the following asymmetric transformation of quinones is a well-known chemistry. Considering the concept of electro-oxidation of hydroquinone followed by asymmetric additions is not new at all, this manuscript doesn't meet this journal's high standard. This manuscript would be more appropriately published in other specialized journals such as *Org. Lett.*

Reviewer #3:

Remarks to the Author:

Comments to the Author

Recommendation: Publish in NC after minor revisions.

Comments:

The development of asymmetric electrochemical catalytic methods for core structure-oriented cross-dehydrogenative coupling (CDC) reactions with high enantioselectivities has attracted significant

attention in recent years. Guo now introduces a copper(I)-catalyzed regiodivergent electrochemical CDC reaction of Schiff bases and commercially available hydroquinones that gives three classes of chiral quaternary amino acid derivatives with excellent regioselectivity, in good to excellent yields and ee values. Interestingly, a concise total synthetic strategy of (-)-fumimycin is also demonstrated. Publication in NC once some issues noted below have been addressed is recommended.

Additional comments:

1: Line 290: the term "Angew. Chem. Int. Ed. 54, 14638–14658 (2017)" can not link to the right paper, it should be "Angew. Chem. Int. Ed. 54, 14638–14658 (2015)" please correct!

2: 2,3-dimethylhydroquinone 2a was available partner to realize the copper(I)-catalyzed regiodivergent electrochemical CDC reaction. Have the authors tried this reaction with 4-aminophenol? It may lead to more interesting results.

Point-by-point response

for

Cu-catalyzed asymmetric regiodivergent electrosynthesis: application in the enantioselective total synthesis of (-)-fumimycin

Tian Xie, Jianming Huang, Juan Li, Lingzi Peng, Jin Song, Chang Guo*

Manuscript number: NCOMMS-23-22922-T

Please find below a list of comments and changes made to the above manuscript in response to reviewers. Further changes made to the manuscript since submission are also listed at the end of this document. **A copy of the revised manuscript, a word document showing tracked changes** made to the manuscript since submission, and **revised Supplementary Information** are also included as part of this revision.

Reply to comments by Reviewer 1

1. Organic electrochemistry has emerged as a powerful tool for organic synthesis. This paper by Guo et al describes an intriguing electrochemical regiodivergent synthesis of valuable amino acid derivatives bearing a quaternary stereocenter in high yields and enantioselectivities. The combination of electrochemistry and Lewis acid catalysis for asymmetric transformation is the key to the achievement of three classes of transformations, based on which asymmetric synthesis of (-)-fumimycin is demonstrated feasible via a concise route. In addition, the mechanistic studies add additional value to the understanding of chemistry. Based on the high challenges of prepared products and the high synthetic value of this methodology, I strongly support the publication of this work in Nature Communications after addressing following minor issues.

Answer: We appreciate reviewer 1 for the favorable comments and helpful suggestions! These comments are greatly valuable and helpful for revising and improving our paper. We have made all the necessary amendments as suggested in our revised manuscript and revised Supplementary Information.

2. There are a few minor points to be considered: In Fig. 3a, such as product 3n, or in Fig. 4 such as product 4m and 4o, I wondered if the authors observed any regioisomers.

Answer: We thank the Reviewer 1 for the important comment. We used Thin Layer Chromatography (TLC) and Nuclear Magnetic Resonance (NMR) techniques to characterize products **3n**, **4m**, and **4o**. However, no discernible isomers were found in any case. To further validate the structure and absolute configuration of **4o**, compound **9** was subjected to X-ray

diffraction analysis (Fig. 6). The results of this analysis provided compelling evidence, confirming both the structure and absolute configuration of **4o**.

3. For Table 1: some abbreviations should be added to the footnote, such as Nap.

Answer: As suggested by Reviewer 1, the abbreviations have been added to the corresponding footnote for Table 1, as “1-Nap, 1-naphthyl. Me, methyl. Bn, benzyl.”.

4. Page 11, change “in the literatures” to “in the literature”.

Answer: We have corrected the error in our revised manuscript.

5. A related reference on asymmetric electrochemical transformation is suggested to be cited: *Chin. J. Org. Chem.* 2020, 40, 3738-3747.

Answer: We have cited the related paper in our revised manuscript (Reference: 39 “39. Wang, X., Xu, X., Wang, Z., Fang, P. & Mei, T. Advances in asymmetric organotransition metal-catalyzed electrochemistry. *Chin. J. Org. Chem.* **40**, 3738–3747 (2020).”).

6. The SI is well prepared and written with good quality of spectra. The eluting solvents for chromatography are recommended to be provided.

Answer: As suggested by Reviewer 1, the eluting solvents utilized in all chromatography procedures have been thoroughly provided in our revised supplementary information.

Reply to comments by Reviewer 2

1. Guo and co-workers report an electrochemical oxidation to generate quinones followed by copper-catalyzed asymmetric addition reactions. Upon alteration of reaction conditions, three types of hydroquinone transformations were realized. My first impression on this manuscript is that it reports a nice work. However, after the literature survey, this referee believes this work doesn't represent an important advance towards asymmetric transformation of hydroquinone.

Answer: We appreciate Reviewer 2 for comments. The routine application of general strategies that enable efficient and economic access to regiodivergent and enantioselective synthesis of valuable amino esters for target-oriented total synthesis remains a critical challenge in organic chemistry. Our manuscript describes a target-oriented, asymmetric regiodivergent electrochemical cross-dehydrogenative coupling reaction for the facile enantioselective total synthesis of (-)-fumimycin. We have introduced a chiral copper catalyst platform for asymmetric regiodivergent electrosynthesis, allowing access to valuable α -quaternary amino esters from racemic substrates. More importantly, the electrochemical anodic oxidation process provides an unique “internal syringe pump” for the controllable release of quinone species which is essential to the stereoselective transformation (Fig. 2d). We believe that our study provides valuable insights into the field of electrochemical asymmetric electrosynthesis and serves as a guide for the further development of powerful and stereocontrolled electrochemical reactions.

2. First, the electrochemical oxidation of hydroquinone to quinone is a well demonstrated chemistry. Even for the electrochemical utilization of in situ generated quinones for asymmetric reactions, many beautiful works have been reported (a seminal work: *Angew. Chem. Int. Ed.* 2009, 10.1002/anie.200904754). Therefore, the chemistry reported herein lacks enough novelty.

Answer: As mentioned by Reviewer 2, the Jørgensen group pioneered the organocatalytic anodic oxidative protocol for the α -arylation of aldehydes with 4-aminophenol to generate substituted anilines. However, the development of asymmetric electrochemical catalytic methods for regiodivergent and core-structure-oriented cross-dehydrogenative coupling reactions remains a challenge that has not been extensively addressed till now. In our research, we aimed to address this challenge by exploring a Lewis acid-catalytic system, coupled with the rational design and development of an electrochemical process, to achieve regiodivergent cross-dehydrogenative coupling reactions of Schiff bases and hydroquinones. Additionally, the synthetic utility of this asymmetric electrochemical methodology enables the total synthesis of (-)-fumimycin. The discovery and identification of asymmetric electrochemical activation methods and concepts are highly desirable, as they inspire further intensive studies and facilitate the development of efficient stereocontrolled electrochemical reactions. Considering the high efficiency, stereoselectivity, and operational simplicity of this transformation, along with the rational design of novel chiral ligands, we anticipate that this method will become a valuable tool in asymmetric synthesis.

3. Second, the asymmetric transformation of quinone has been realized by many groups (ref. 13-21 and others). In the previous reports, excellent enantioselectivities could be achieved. However, in the present work, most of the ee values are below 94%. Overall, the main achievement of this work is the use of electrons as chemical oxidant alternatives to generate quinones from hydroquinones, while the following asymmetric transformation of quinones is a well-known chemistry. Considering the concept of electro-oxidation of hydroquinone followed by asymmetric additions is not new at all, this manuscript doesn't meet this journal's high standard. This manuscript would be more appropriately published in other specialized journals such as *Org. Lett.*.

Answer: We appreciate reviewer 2 for the comments. In our study, we aimed to demonstrate the unique capabilities of organic electrochemistry in achieving regiodivergent electrochemical CDC reactions of Schiff bases and hydroquinone derivatives. Although asymmetric transformations of quinone have been reported previously, our work focuses on the development of a copper-catalyzed, regiodivergent electrochemical CDC reaction that allows for the efficient synthesis of chiral quaternary amino acid derivatives with excellent stereocontrol. This allows for the efficient assembly of complex scaffolds and the generation of three different classes of chiral quaternary amino acid derivatives with excellent stereocontrol. Additionally, we found that by implementing a controlled release of reactive quinones from the in situ anodic oxidation process of hydroquinones using an "internal syringe pump" protocol, high levels of reaction efficiency and enantiocontrol can be achieved. This novel approach demonstrates the tunability and precise control over redox transformations offered by organic electrochemistry. Moreover, the synthetic utility of our strategy is exemplified through the asymmetric total synthesis of (-)-fumimycin, highlighting the practical application and potential of this methodology.

Reply to comments by Reviewer 3

- The development of asymmetric electrochemical catalytic methods for core structure-oriented cross-dehydrogenative coupling (CDC) reactions with high enantioselectivities has attracted significant attention in recent years. Guo now introduces a copper(I)-catalyzed regiodivergent electrochemical CDC reaction of Schiff bases and commercially available hydroquinones that gives three classes of chiral quaternary amino acid derivatives with excellent regioselectivity, in good to excellent yields and ee values. Interestingly, a concise total synthetic strategy of (-)-fumimycin is also demonstrated. Publication in NC once some issues noted below have been addressed is recommended.

Answer: We appreciate reviewer 3 for the favorable comments and helpful suggestions! These comments are greatly valuable and helpful for revising and improving our paper. We have made all the necessary amendments as suggested in our revised manuscript and revised Supplementary Information.

- Line 290: the term “Angew. Chem. Int. Ed. 54, 14638–14658 (2017)” can not link to the right paper, it should be “Angew. Chem. Int. Ed. 54, 14638–14658 (2015)” please correct!

Answer: We have corrected the error in our revised manuscript.

- 2,3-dimethylhydroquinone 2a was available partner to realize the copper(I)-catalyzed regiodivergent electrochemical CDC reaction. Have the authors tried this reaction with 4-aminophenol? It may lead to more interesting results.

Answer: We thank the reviewer 3 for the important comment and helpful suggestions regarding the 4-aminophenols as reaction partners. Those comments are greatly valuable and helpful for revising and improving our paper. We carried out electrochemical reactions of racemic ketimine esters with various 4-aminophenols, including N-(4-hydroxyphenyl)-4-methylbenzenesulfonamide (**2i**), 4-(methylamino)phenol (**2j**), and 4-amino-2,3-dimethylphenol (**2k**). The following outcomes have been included in our revised Supplementary Information (Page S40, Figure S4).

As depicted in Figure S4 (Page S40), the utilization of different 4-aminophenol partners exhibited a significant impact on reactivity (a-c). No reaction was observed when employing **2i** and **2j** as substrates. Encouragingly, **2k** was found to be an appropriate reaction partner. The desired product **12** with a yield of 56% and an enantiomeric excess (e.e.) of 38% was obtained while utilizing (*S,S*_p)-**L4** as the chiral ligand. Several chiral PHOX ligands were evaluated to promote the reaction, and the desired product **12** could be obtained in good yield and enantioselectivity (62% yield, 86% e.e.) with (*S,S*_p)-**L6**.

Methyl (S,E)-2-((2,3-dimethyl-4-oxocyclohexa-2,5-dien-1-ylidene)amino)-5-phenyl-3,4-dihydro-2H-pyrrole-2-carboxylate (12)

The title compound was prepared according to the general procedure D using **1a** (0.15 mmol), **2k** (0.225 mmol) and (*S,S*_p)-**L6** as ligand at 10 °C. The crude reaction mixture was purified by flash column chromatography (petroleum ether/ethyl acetate, 6:1) to afford the title compound as a yellow oil (62%). ¹H NMR (600 MHz, CDCl₃) δ 7.93 – 7.89 (m, 2H), 7.68 (d, *J* = 10.3 Hz, 1H), 7.50 – 7.45 (m, 1H), 7.45 – 7.39 (m, 2H), 6.45 (d, *J* = 10.3 Hz, 1H), 3.75 (s, 3H), 3.25 – 3.12 (m, 2H), 3.04 – 2.96 (m, 1H), 2.55 – 2.47 (m, 1H), 2.19 (s, 3H), 2.01 (s, 3H). ¹³C NMR (151 MHz, CDCl₃) δ 187.38, 176.24, 172.22, 160.17, 145.41, 136.75, 133.55, 131.65, 131.10, 130.86, 128.65, 128.53, 98.23, 53.20, 37.52, 35.03, 14.40, 12.26. ESI-MS: calculated [C₂₀H₂₀N₂O₃ + H]⁺: 337.1547, found: 337.1551. [α]_D²⁰ = -198.5 (c = 0.18, CH₂Cl₂). The product was analyzed by HPLC to determine the enantiomeric excess: 86% e.e. (CHIRALPAK AD-H, hexane/*i*-PrOH = 70/30, detector: 268 nm, T = 25 °C, flow rate: 1 mL/min), t₁(major) = 5.10 min, t₂(minor) = 6.06 min.

¹H NMR of 12

¹³C NMR of 12

Rac-12

SAMPLE INFORMATION			
Sample Name:	xt-5-109-rac-30%-AD	Acquired By:	System
Sample Type:	Unknown	Sample Set Name:	
Vial:	69	Acq. Method Set:	30%qb
Injection #:	1	Processing Method:	xt 5 109 rac
Injection Volume:	20.00 ul	Channel Name:	268.0nm
Run Time:	60.0 Minutes	Proc. Chnl. Descr.:	2998 PDA 268.0 nm (2998)
Date Acquired:	8/9/2023 10:17:03 AM CST		
Date Processed:	8/9/2023 10:32:24 AM CST		

	RT	Area	% Area	Height
1	5.095	2837833	48.70	361083
2	6.048	2989318	51.30	319848

Asy-12

SAMPLE INFORMATION			
Sample Name:	xt-5-109-2-30%-AD	Acquired By:	System
Sample Type:	Unknown	Sample Set Name:	
Vial:	37	Acq. Method Set:	30%qb
Injection #:	1	Processing Method:	xt 5 109 2
Injection Volume:	20.00 ul	Channel Name:	268.0nm
Run Time:	60.0 Minutes	Proc. Chnl. Descr.:	2998 PDA 268.0 nm (2998)
Date Acquired:	8/9/2023 10:03:26 AM CST		
Date Processed:	8/9/2023 10:27:54 AM CST		

	RT	Area	% Area	Height
1	5.102	12689574	93.07	1608187
2	6.060	944169	6.93	102337

Reviewers' Comments:

Reviewer #3:

Remarks to the Author:

I have read the author's responses carefully.

Guo and co-workers have answered the comments of Reviewer 1 and Me very well.

I think this work is worth to be published in Nature Communications.